# Microbial Profile Antibacterial Properties and Chemical Composition of Raw Donkey Milk

**DOI:** 10.3390/ani10112001

**Published:** 2020-10-30

**Authors:** Theofilos Massouras, Nefeli Bitsi, Spiros Paramithiotis, Eugenia Manolopoulou, Eleftherios H. Drosinos, Kostas A. Triantaphyllopoulos

**Affiliations:** 1Department of Food Science and Human Nutrition, Laboratory of Dairy Science and Technology, Agricultural University of Athens, 75 Iera Odos St., 11855 Athens, Greece; nbitsi@yahoo.com (N.B.); mae@aua.gr (E.M.); 2Department of Food Science and Human Nutrition, Laboratory of Food Quality Control and Hygiene, Agricultural University of Athens, 75 Iera Odos St., 11855 Athens, Greece; sdp@aua.gr (S.P.); ehd@aua.gr (E.H.D.); 3Department of Biotechnology, School of Applied Biology and Biotechnology, Agricultural University of Athens, 75 Iera Odos St., 11855 Athens, Greece; ktrianta@aua.gr

**Keywords:** donkey milk, RAPD-PCR, 16S rDNA sequencing, antimicrobial, lactic acid bacteria, chemical composition

## Abstract

**Simple Summary:**

Apart from the economically exploited ruminants, alternative milk producing species, such as donkey, camel, reindeer, llama and yak have made an economic impact in many countries. However, their potential in human health and nutrition remains underexploited. In Europe, donkey milk is mainly produced in small-scale farms. However, consumer interest in donkey milk increases and is gaining international acceptance. This product is used as a substitute for human nutrition and susceptible consumer groups, i.e., infants and children who suffer from cow’s milk protein intolerance. The current study examines the donkey milk microbiota. In more detail, the findings have shown the presence of lactic acid bacteria in donkey milk, which is an important source for selecting starter cultures. Some pathogenic bacteria have been also identified. The European Union has adopted regulations (852/2004/EC) for donkey milk consumption, in order to achieve a high level of protection for human health. The conducted research focuses on the microbiological profile, antimicrobial properties, and chemical composition of raw donkey milk, in order to provide useful information and improve expertise on the nutritional value and hygiene quality of donkey milk, from two representative Greek breeds.

**Abstract:**

The human interest in donkey milk is growing due to its nutritional, functional properties and excellent microbiological quality according to published reports. However, more research needs to be conducted to assess the above variables from various breeds. In the present study, milk samples were collected from 17 Cypriot and six Arcadian healthy Greek donkeys. The microbiological quality, somatic cell counts (SCC), chemical composition analysis, and antimicrobial activity of the samples was assessed. In addition, clustering and identification of the bacterial composition was performed by RAPD-PCR and 16S rDNA sequencing, respectively. The good microbiological quality of the samples as estimated by the total aerobic mesophilic and psychrotrophic counts, which ranged from 2.18 to 2.71 log CFU/mL and from 1.48 to 2.37 log CFU/mL, respectively, was also verified. SCC were below 4.4 log CFU/mL. However, potential pathogenic species of *Staphylococcus aureus, Bacillus cereus*, and *Clostridium* spp. were enumerated in the milk of both breeds. The gross chemical composition showed mean values for fat, protein, and lactose from 0.82% to 1.24%, 1.22% to 1.87%, and 6.01% to 6.78%, respectively. All milk samples exhibited an antimicrobial activity against *St. haemolyticus* and *Listeria monocytogenes*, although quality control measures should be taken for health and safety prior to human consumption.

## 1. Introduction

An increased interest in donkey milk has occurred over the last decade mainly due to its nutritional composition and functional properties. Indeed, donkey milk has been reported to contain less total protein and fat and more lactose than bovine milk [1]. Moreover, it is richer in mono- and poly-unsaturated fatty acids, vitamins C and B_12_, calcium and phosphorus, and contains a comparable amount of essential amino acids [2,3,4,5,6]. As far as the functional properties are concerned, they are summarized into the hypoallergenic potential, mostly due to the low casein content and the immunomodulation activity [7,8]. 

The excellent microbiological quality of raw donkey milk has been repeatedly exhibited. The total aerobic mesophilic count reported is usually below 4 log CFU/mL [9,10,11] with only sporadic exceptions [12]. In addition, the occurrence of foodborne pathogens such as *Listeria monocytogenes*, *Salmonella* serovars, and *Escherichia coli* O157 are yet to be reported [13]. This excellent safety record has been attributed to the synergistic action of lactoferrin, lysozyme, immunoglobulins, and fatty acids [14], as well as anatomical reasons associated to the size and position of the udder [6].

Several high-resolution molecular typing methods have been developed and effectively applied for the assessment of lactic acid bacteria biodiversity. Among them, random amplified polymorphic DNA (RAPD) has been repeatedly used for clustering and differentiation of lactic acid bacteria isolates offering simplicity, speed, cost-efficiency, and molecular taxonomy attributes, that enable us to distinguish members of microbiota at the sub-species level [15,16,17,18].

The aim of the present study was to assess the microbiological and antimicrobial characteristics of raw donkey milk of two donkey breeds, reared in a designated farm at Argos, Greece.

## 2. Materials and Methods

### 2.1. Animals and Sampling

This study was carried out in a donkey farm located in the region of Argos, Peloponnese for a period of 5 months (November–March). Milk samples were collected from 23 clinically healthy donkeys 17 Cypriot (population C) and six Arcadian (population A) breeds. In the beginning of the experiment, the animals were not all at the same lactation period. Specifically, among the 17 animals of the Cypriot breed (C1–C5), two of them were in the 30th day after delivery (C1), four in the 60th day (C2), four in the 90th day (C3), two in the 120th (C4), and five in the 150th of lactation (C5), while the six animals of the Arcadian breed (A1) were in the 30th day after delivery of the newborns. All donkeys in the experiment were milked manually twice daily (6 a.m. and 6 p.m.). Individual fresh milk samples, pooled from the morning and evening milking, were refrigerated immediately at 4 °C until the microbiological and somatic cell count (SCC) assessment, as well as chemical analysis, which were performed the next day. In total, 115 samples were collected for the experiment, spanning over the entire lactation period from the 30th to the 270th day (Table 1).

### 2.2. Chemical Analyses

Fresh milk samples were analyzed for crude protein, fat, lactose, solids non-fat (SNF), and total solids (TS) by the Fourier transform infrared (FTIR) analysis using the MilkoScan FT 6000 (Foss Electric, Hillerød, Denmark), previously standardized for donkey milk according to the Joint Standard ISO 9622/IDF141:2013 [19]. Ash content was determined after incineration in a muffle furnace at 530 °C [20].

### 2.3. Microbiological and Somatic Cell Count Analyses

Samples, 10 mL, were homogenized with 90 mL of sterile Ringer solution using a stomacher apparatus (Seward Medical, London, UK); appropriate serial dilutions were prepared in the same diluent. Plating and surface spreading techniques were performed by mixing 1 mL of the diluted sample with molten media or by spreading 0.1 mL of the diluted sample on the surface of the media, respectively. In all cases, duplicate plates were prepared for culturing the microorganisms. The total aerobic mesophilic count was estimated by spreading on Plate Count Agar (Oxoid, Hampshire, UK), followed by incubation at 30 °C for 72 h. Enumeration of lactic acid bacteria (LAB) was carried out by plating on MRS agar (Oxoid), followed by anaerobic incubation (GasPak, BBL, Cockeysville, MD, USA) at 30 °C for 48 h (ISO, 1998) [21]. *Lactococcus* spp. and *Enterococcus* spp. populations were determined by plating on M17 (Lab M, Lancashire, UK) and Kanamycin Aesculin Azide agar (KAA) (Lab M, Lancashire, UK), after incubation at 30 °C for 48 h and 37 °C for 24 h, respectively. *Micrococcus* spp. was enumerated by plating on Mannitol Salt Agar (Oxoid) supplemented with cycloheximide (100 µg/mL; Sigma-Aldrich, St. Louis, MO, USA), followed by incubation at 30 °C for 72 h. The psychrotrophic bacteria count was estimated by spreading on milk PCA and after incubation at 7 °C for 10 days. Enumeration of coliforms and *Clostridium* spp. was carried out by pouring on Violet Red Bile Agar (Oxoid) and Reinforced Clostridial agar (Lab M), followed by incubation at 37 °C for 24 h [22] and 30 °C for 72 h, respectively. Yeasts/molds and *Bacillus cereus* populations were estimated by spreading on Yeast Glucose Chloramphenicol (Merck, Darmstadt, Germany) and *Bacillus* agar base medium (OXOID), followed by incubation at 30 °C for 72 h [23] and 37 °C for 24 h [24], respectively.

Somatic cell counts (SCC) were estimated by following the procedure ISO 13366-2:2006 [25] using the Fossomatic 5000 (Foss-Εlectric, Hillerød, Denmark).

### 2.4. Bacterial Isolation and Identification 

From each sample, the grown colonies on MRS and M17 agars, were selected according to the representative sampling scheme proposed by Harrigan and McCance (1978) and isolated by successive subculturing on the same media [26]. Isolates were phenotypically characterized on the basis of their ability to grow at 10, 15, and 45 °C, 6.5% (*w*/*v*) NaCl, pH 9.6, gas production from glucose along with their Gram stain and catalase reaction, according to Drosinos et al. (2007) [27]. 

Genotype clustering of the isolates was performed by random amplified polymorphic DNA-polymerase chain reaction (RAPD-PCR). For this purpose, the extraction of DNA was performed according to Paramithiotis et al. (2010) [28] and RAPD-PCR analysis was followed by using M13 as the only primer. In more detail, the PCR was performed in 25 µL volume containing 0.2 mM dNTPs (Peqlab, Erlangen, Germany), 2.5 mM MgCl_2_, 4 µΜ primer M13 (5′-GAG GGT GGC GGT TCT-3′), and 2 U *Taq* polymerase (Biotools, Madrid, Spain). Thermocycling conditions were as follows: Initial denaturation at 95 °C for 2 min, 35 cycles of 95 °C for 1 min, 38 °C for 1 min ramp to 72 °C at 0.6 °C s^−1^, 72 °C for 2 min, and a final extension step at 72 °C for 10 min. DNA fragments were separated by electrophoresis in 1.5% agarose gel in 1.0× TAE at 100 V for 1.5 h, visualized by ethidium bromide staining and photographed using a GelDoc system (Bio-Rad, Hercules, CA, USA). Conversion, normalization, and further analysis were performed with the Bionumerics software version 6.1 (Applied Maths NV, Sint-Martens-Latem, Belgium) using the Pearson’s coefficient and UPGMA cluster analysis. One to three representative strains from each cluster were subjected to 16S-rRNA gene sequencing, according to Cocolin et al. (2004) [29], for taxonomic assignment. PCR amplification products were subsequently bi-directionally sequenced. Finally, the sequences were further aligned with those found in GenBank using the BLAST program to determine the closest known relatives.

### 2.5. Antimicrobial Activity Assay

Determination of the antimicrobial activity was performed with the well diffusion assay (WDA). More accurately, in freshly prepared lawns of overnight growth of the indicator strains in BHI agar (Lab M), wells were aseptically punched. Then, 20 µL of each sample were added in the wells. Incubation was carried out at 37 °C for 24 h. Inhibition of the indicator strain’s growth around the wells suggested the presence of antimicrobial activity of the used sample. The indicator strains used were, *Staphylococcus aureus* (eight strains), *St. haemolyticus* (three strains), *Enterococcus faecium* (five strains), and *Escherichia coli* (one strain) isolated from fermented meat products, *Salmonella* serovar Typhimurium isolated from poultry, and serovars Agona, Infantis, Reading, Emek, Seftenberg, Livingstone, Putten (two strains), and Cubana isolated from feeds, *Listeria monocytogenes* isolated from strawberries (five strains) and cured meat products (four strains), as well as *E. coli* ATCC 25922, *E. coli* O157:H7 NCTC 12079, NCTC 13125, NCTC 13127, *St. aureus* ATCC 6538, and NCBF 1499 [27,30,31].

### 2.6. Statistical Analysis

One-way analysis of variance (ANOVA) (MS Excel, 2010) was used to statistically assess the differences between the microbial population dynamics, at the significance level of *p* < 0.05. The breed factor was not included in the analysis because of small numbers in each of the two breed groups.

## 3. Results 

Τhe chemical composition of the donkey milk samples is presented in Table 2. The mean value for fat, protein, lactose, non-fat dry matter, total solids, and ash ranged from 0.82% to 1.24%, 1.22% to 1.87%, 6.01% to 6.78%, 7.23% to 8.65%, 8.37% to 9.50%, and 0.343% to 0.438%, respectively. No statistically significant (*p* < 0.05) differences were observed between the samples, either between breed C or between the breeds C and A.

The microbiological profile of the donkey milk samples is shown in Table 3. Total mesophilic aerobic and psychrotrophic bacteria exhibited an average value of 2.18 to 2.71 log CFU/mL and from 1.48 to 2.37 log CFU/mL, respectively. All the genera assessed ranged within or below the latter values. All milk samples exhibited a comparable microbiological quality and no statistically significant (*p* < 0.05) differences were observed. Table 3 shows also the somatic cell counts of the donkey milk samples, which is considered an indicator of animal health and milk quality. SCC values in this study were below 100,000 cells/mL. No statistically significant differences (*p* < 0.05) were observed between the groups of samples at the different stages of lactation (Table 3). 

Importantly, concerning the antimicrobial activity of the donkey milk samples, growth inhibition was observed only for two indicator strains, namely *St. haemolyticus* LQC 5021 and *L. monocytogenes* LQC 15017.

A total of 85 strains were isolated from the donkey milk samples and classified into two major groups consisting of 78 Gram positive, catalase negative and 7 Gram positive, catalase positive strains that were also coagulase positive. The catalase negative samples were further subdivided into four groups. Thus, 24 strains were hetero-fermentative bacilli able to grow at 15 but not 45 °C, 6.5% NaCl and pH 9.6; three strains were hetero-fermentative cocci unable to grow at 45 °C and 6.5% NaCl; 27 strains were homo-fermentative cocci able to grow at 10 and 45 °C, as well as 6.5% NaCl and pH 9.6; 20 strains were homo-fermentative cocci and able to grow at 10 °C but not at 45 °C as well as 6.5% NaCl and pH 9.6. Finally, four strains were homo-fermentative cocci able to grow at 45 °C, but not 6.5% NaCl and pH 9.6. On the basis of origin, colony and cell morphology as well as biochemical tests, 49 isolates were selected for genotype clustering and identification. Genotype clustering was performed by RAPD-PCR using M13 as a primer; in Figure 1 the cluster analysis of RAPD-PCR patterns of the bacterial isolates is demonstrated. The isolates under study were effectively separated into several clusters. One to three representative strains from each cluster were subjected to sequencing of the V1–V3 region of 16S-rRNA gene and the resulting phylogenetic position is presented in Table 4, while the distribution of the bacterial isolates in the milk samples is presented in Table 5. A total of 15 isolates were identified as *Levilactobacillus brevis* that were present in all samples of Cypriot breed; on the contrary, it was not isolated from the Arcadian breed. *Enterococcus* spp. was also widespread among all the samples. Specifically, *Ec. mundtii* was isolated from three samples of the Cypriot and the Arcadian breed, *Ec. durans* and *Ec. faecium* from two samples of the Cypriot breed but only the former from the Arcadian breed, whereas *Ec. faecalis* was only present in the sample of the Arcadian breed. *Staphylococcus aureus* was isolated from two samples of the Cypriot breed and the Arcadian breed, *Lactococcus lactis* and *Streptococcus macedonicus* from two samples of the Cypriot breed. Finally, *Leuconostoc mesenteroides* was isolated from only one sample from the Cypriot breed. Based on the number of the isolated colonies from each sample, *Lc*. *lactis* seems to prevail in C1 and C4 samples, *Lb. brevis* in C2 and C5 samples, a consortium consisting of *Lb*. *brevis*, *Enterococcus* spp., and *Str*. *macedonicus* in C3 samples, and a group mainly consisted of *Enterococcus* spp. in A1 samples.

## 4. Discussion

Based on our findings, the quantitative assessment of the constituents of the studied samples (Table 2) was in agreement with previous reports in the literature [1,32]. In relation to milk composition, fat content was reported to vary from 0. 8 to 2.05 g/100 mL, protein content from 0.85 to 2.08 g/100 mL, lactose content from 3.54 to 8.46 and total solids from 8.8 to 11.7 g/100 mL [1,12,33]. This variation has been assigned to breed, age, feeding regime, lactation stage, and season, as well as milking strategy and technique [1,10,34]. In this regard, concerning the Cypriot breed, no statistically significant differences were observed, possibly as a result of variability between the breeds.

From a microbiological perspective, the total aerobic mesophilic count of 2.18–2.71 log CFU/mL verified the good microbiological quality of the donkey milk [12,35,36] in the examined samples. This has previously been assigned to anatomic reasons associated to the size and position of the udder [6], as well as to the presence of antimicrobials [13]. 

Concerning the presence of foodborne pathogens, the occurrence of *B. cereus* and *Staphylococcus* spp. is frequently reported in the literature [10,11,12,37,38], however, this is the first time that the presence of *Clostridium* spp. was reported. The occurrence of *Clostridium* spp. and *Bacillus cereus* in the donkey milk indicates the potential presence of pathogenic microorganisms in the milk. Thus, there is a need for pasteurization of the milk and strict hygienic measures during production and handling of the milk. Quality control measures should be put in place to ensure that donkey milk sold in the market is safe for human consumption. Under these conditions, the Greek National Legislation (Ministerial Decision 314/15074/2014, Government Gazette 363/Β/17/2/2014), requires that the production and processing of equine milk is to be complied with the criteria of the EC regulations 2073/2005, 852/2004, and 853/2004.

The antimicrobial activity of donkey milk has been extensively studied; it has been reported against a series of foodborne pathogens including *B. cereus*, *Enterococcus faecalis*, *Escherichia coli*, *Listeria monocytogenes*, *Salmonella enterica* (formerly *S. choleraesuis*), *S. enteritidis*, *S. livingstone*, *S. typhimurium*, *Shigella dysenteriae,* and *Staphylococcus aureus* [35,39,40,41,42] and has been attributed to the synergistic action of lactoferrin, lysozyme, immunoglobulins, and fatty acids [14]. In the present study, it was only demonstrated against one *St. haemolyticus* and one *L. monocytogenes* strain, indicating the strain-dependent character of this property.

The approach used in the current study for biotyping and identification of the isolates, i.e., clustering of the isolates through the use of a genotypic technique and taxonomic assignment of a representative number of strains through sequencing of V1–V3 region of 16s-rRNA gene has been extensively applied [16,43,44,45]. Among molecular techniques, RAPD is considered a fast method widely used for the characterization of bacteria and specifically for the characterization of lactic acid bacteria in dairy products [46]. 

As far as the approach used is concerned, the current technique is prone to minor reproducibility issues that arise from nearly every experimental parameter implicated with the analysis [47]. However, in the present study, no such problems have been encountered due to the precise and careful standardization. Another feature that emerges from this technique is that isolates belonging to the same species are often clustered separately [16,48]. The latter was also the case in the present study and may be explained when clonality of the microorganisms was assessed, as well as when the underlying principle of this technique was considered. Moreover, principally clonal populations essentially lead to very similar genotype profiles. The latter was the case of the *St. aureus* assessed in the present study; isolates from three samples were assembled in one cluster, revealing the principally clonal nature of this microorganism, in accordance with the literature [49]. As far as the underlying principle of this technique is concerned, in respect to the randomness of the amplified fragments, the correlation coefficient calculated after the analysis of randomly generated genotype profiles may not accurately depict the extent of the genetic relatedness and thus the relationships inferred may be misleading. To this end, the inability of V1–V3 region of 16S rRNA sequencing to distinguish between closely related species, such as *Ec. mundtii*, *Ec. durans,* and *Ec. faecium* should be taken into consideration. In addition, the need to combine phenotype and genotype methodology is especially evident, for example, to discriminate within the *Lactococcus lactis* species, as strains within the *cremoris* genotype that may have the *lactis* phenotype [50]; thus, this could further certify the taxonomic origin of the microorganism.

As far as the composition of the lactic acid microbiota is concerned, significant differences between the results obtained in the present study and the ones reported in the literature, were observed. Carminatti et al. (2014) has previously reported the dominance of *Enterococcus* spp. and *Streptococcus* spp. in donkey milk samples [51]. More accurately, *Ec. faecium*, *Ec. faecalis,* and *Str. macedonicus* dominated the microbiota with an infrequent presence of *Ec. gilvus*, *Ec. casseliflavus*, *Str. equinus*/*bovis*, *Str. equi,* and *Str. criceti*. Similar results were also reported by Aspri et al. (2017) [36]; *Enterococcus* spp. and *Streptococcus* spp. seemed to prevail in the examined samples and only a small percentage of the microecosystem consisted of *Lactobacillus* spp. and *Leuconostoc* spp. In the present study, there was only a partial agreement with the aforementioned studies. More accurately, the microbial load for *Enterococcus* spp., *Lc. lactis,* and *Lb. brevis* was the most prominent and important member of the microecosystem, including the infrequent presence of *Streptococcus* spp. and *Ln. mesenteroides*.

## 5. Conclusions

In the present study, microbe profiling of donkey milk has been investigated and the taxonomic diversity among LAB strains has been revealed. Dominant LAB strains, such as *Enterococcus* spp., *Lc. lactis,* and *Lb. brevis* were identified. 

Our data demonstrated that, albeit at a low bacterial content in donkey milk, the potential presence of pathogenic species is not excluded, which underlines the importance of heat treatment prior to human consumption. The presence of *Clostridium* spp. and *B. cereus* in raw donkey milk indicates the potential presence of pathogenic microorganisms in the milk. Therefore, there is a need for pasteurization and strict hygienic measures upon milk production and handling. Quality control measures should be put in place to ensure that the donkey milk sold in the market is safe for human consumption.

## Figures and Tables

**Figure 1 animals-10-02001-f001:**
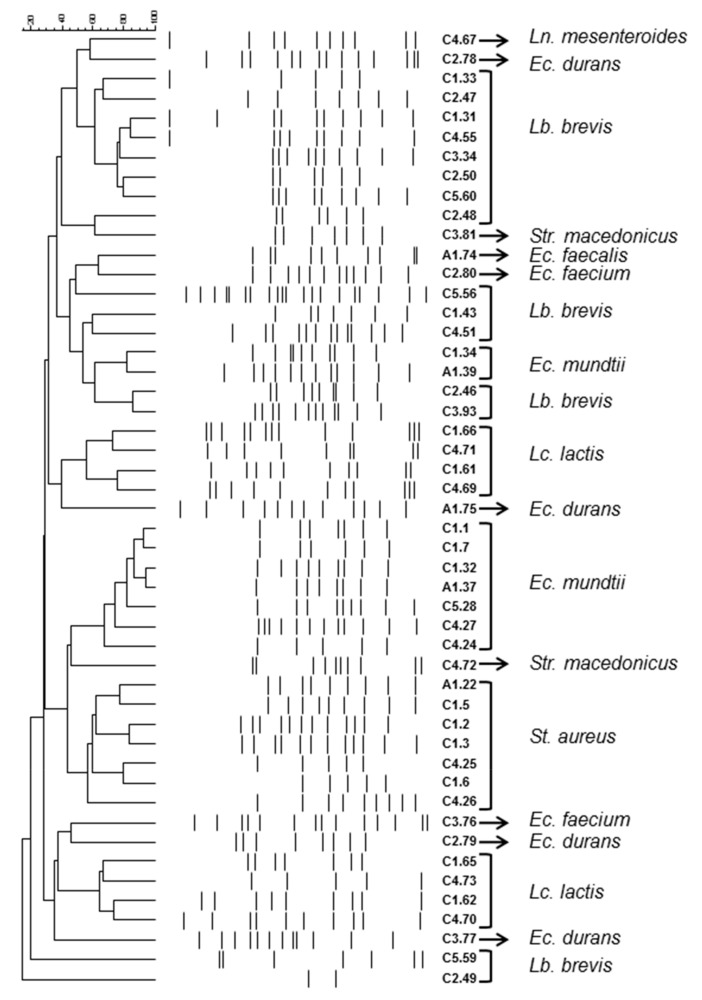
Cluster analysis of random amplified polymorphic DNA-polymerase chain reaction (RAPD-PCR) patterns of the bacterial isolates. Distance is indicated by the mean correlation coefficient (r%) and clustering was performed by unweighted pair group method with arithmetic mean (UPGMA) analysis. Each isolate is named after the origin (C1 to C5; A1) followed by the strain number. Taxonomic affiliation was based on sequencing of V1–V3 region of 16S rRNA gene. *Ec*.: *Enterococcus*; *Lb*.: *Levilactobacillus*; *Lc*.: *Lactococcus*; *Ln*.: *Leuconostoc*; *St*.: *Staphylococcus*; *Str*.: *Streptococcus.*

**Table 1 animals-10-02001-t001:** The donkey milk sampling during the study period.

Breed	Animals/Lactation Period	Sampling Month					No. of Samples
		NOV	DEC	JAN	FEB	MAR	(*n*)
		Days					
C1	2	30	60	90	120	150	10
C2	4	60	90	120	150	180	20
C3	4	90	120	150	180	210	20
C4	2	120	150	180	210	240	10
C5	5	150	180	210	240	270	25
A1	6	30	60	90	120	150	30
Total	23						115

Breed: Cypriot (C1–C5), Argos (A1).

**Table 2 animals-10-02001-t002:** Chemical composition of the donkey milk samples.

	C1 (n = 10)	C2 (n = 20)	C3 (n = 20)	C4 (n = 10)	C5 (n = 25)	A1 (n = 30)	*p <* 0.05
Fat (%)	1.24 (0.6)	0.85 (0.3)	1.14 (0.3)	1.20 (0.3)	0.82 (0.1)	1.05 (0.3)	ns
Protein (%)	1.36 (0.2)	1.87 (0.2)	1.30 (0.3)	1.22 (0.2)	1.27 (0.3)	1.27 (0.1)	ns
Lactose (%)	6.18 (0.9)	6.54 (0.4)	6.78 (1.5)	6.07 (0.4)	6.01 (0.3)	6.13 (0.3)	ns
Non-fat dry matter content (%)	7.8 (0.5)	8.65 (0.2)	8.04 (1.4)	7.23 (0.3)	7.31 (0.5)	7.35 (0.2)	ns
Total solids (%)	9.03 (0.5)	9.50 (0.4)	9.18 (1.9)	8.73 (0.5)	8.37 (0.4)	8.69 (0.4)	ns
Ash (%)	0.413 (0.09)	0.385 (0.012)	0.428 (0.011)	0.343 (0.016)	0.372 (0.025)	0.438 (0.080)	ns

The average values are presented. Standard deviation is given in parenthesis. Within a row, no significant differences (ns) were detected (*p* < 0.05).

**Table 3 animals-10-02001-t003:** Microbiological profile (in log CFU Ml^−1^) and somatic cell counts (in log cells mL^−1^ of the donkey milk samples.

	C1	C2	C3	C4	C5	A1	*p* < 0.05
TAMC	2.18 (0.9)	2.38 (0.4)	2.54 (0.4)	2.55 (0.1)	2.71 (0.5)	2.37 (0.4)	ns
LAB	1.67 (0.9)	0.99 (0.1)	1.19 (0.8)	1.38 (0.8)	1.45 (0.2)	1.18 (0.8)	ns
Lactococci	2.56 (0.2)	1.95 (0.6)	1.84 (0.2)	1.76 (0.3)	2.35 (0.1)	1.83 (0.2)	ns
Enterococci	0.58 (0.7)	0.50 (0.9)	0.74 (1.4)	0.89 (0.6)	0.77 (0.9)	0.22 (0.2)	ns
Psychrotrophic bacteria	1.48 (1.4)	2.34 (0.9)	2.37 (0.8)	1.48 (0.9)	2.36 (0.8)	1.83 (0.8)	ns
Micrococci	2.86 (0.5)	1.85 (1.2)	2.18 (0.9)	2.45 (0.3)	2.70 (0.9)	2.33 (0.5)	ns
Coliforms	1.86 (0.6)	1.29 (1.2)	2.18 (0.9)	1.92 (0.8)	1.50 (0.7)	1.08 (0.1)	ns
Υeasts/moulds	2.56 (0.3)	2.95 (1.2)	1.86 (0.6)	2.42 (0.1)	2.54 (0.8)	2.56 (0.3)	ns
*Clostridium* spp.	1.23 (0.8)	1.18 (0.4)	2.18 (0.9)	1.36 (0.1)	1.99 (0.8)	1.13 (1.2)	ns
*Bacillus cereus*	1.53 (1.3)	1.73 (1.4)	2.18 (0.9)	1.15 (0.8)	1.32 (0.9)	1.12 (0.9)	ns
SCC	4.40 (0.09)	4.32 (0.10)	4.05 (0.20)	4.22 (0.08)	3.92 (0.12)	3.79 (0.31)	ns

TAMC: Total Aerobic Mesophilic Count; LAB: Lactic Acid Bacteria; SCC: Somatic Cell Count. The average values are presented. Standard deviation is given in parenthesis. Within a row, no significant differences (ns) were detected (*p* < 0.05).

**Table 4 animals-10-02001-t004:** Phylogenetic position of selected strains based on sequencing of V1–V3 region of 16S rRNA gene.

Isolate Number ^1^	Closest Relative	Identity (%)	Accession Number
C1.2	*St. aureus*	99	CP009361
C4.24	*Ec. mundtii*	94	KM005159
C1.32	*Ec. mundtii*	99	KM005159
C1.33	*Lb. brevis*	99	KC713915
A1.39	*Ec. mundtii*	99	KM005159
C2.46	*Lb. brevis*	98	KC713915
C2.48	*Lb. brevis*	97	KJ994501
C2.49	*Lb. brevis*	99	KJ994501
C2.50	*Lb. brevis*	99	KC713915
C4.51	*Lb. brevis*	97	KJ994501
C5.56	*Lb. brevis*	99	KC713915
C5.59	*Lb. brevis*	99	KC713915
C4.67	*Ln. mesenteroides*	99	JQ800447
C4.70	*Lc. lactis*	99	KJ958440
C4.71	*Lc. lactis*	99	KJ958440
C4.72	*Str. macedonicus*	98	AF459431
A1.74	*Ec. faecalis*	99	HF558530
A1.75	*Ec. durans*	99	KJ958437
C3.76	*Ec. faecium*	99	KJ832070
C3.77	*Ec. durans*	99	KJ702532
C2.78	*Ec. durans*	100	KF250785
C2.79	*Ec. durans*	99	KF250827
C2.80	*Ec. faecium*	99	KJ958432
C3.81	*Str. macedonicus*	99	JX850868

^1^ The number of each isolate consists of the number of the sample followed by the number of the strain. *Ec*.: *Enterococcus*; *Lb*.: *Levilactobacillus*; *Lc*.: *Lactococcus*; *Ln*.: *Leuconostoc*; *St*.: *Staphylococcus*; *Str*.: *Streptococcus.*

**Table 5 animals-10-02001-t005:** Distribution of the bacterial isolates in the studied donkey milk samples.

Bacteria Species	C1	C2	C3	C4	C5	A1
*Ec. durans*		2 (3)	1 (2)			1 (2)
*Ec. faecalis*						1 (3)
*Ec. faecium*		1 (2)	1 (1)			
*Ec. mundtii*	4 (6)			2 (3)	1 (2)	2 (3)
*Lb. brevis*	3 (5)	5 (7)	2 (3)	2 (4)	3 (5)	
*Lc. lactis*	4 (11)			4 (9)		
*Ln. mesenteroides*				1 (3)		
*St. aureus*	4 (4)			2 (2)		1 (1)
*Str. macedonicus*			1 (2)	1 (2)		

The number outside the parenthesis indicates the number of the identified strains. The number inside the parenthesis indicates the number of the strains allocated to the species after integration of the respective phenotype properties. *Ec*.: *Enterococcus*; *Lb*.: *Levilactobacillus*; *Lc*.: *Lactococcus*; *Ln*.: *Leuconostoc*; *St*.: *Staphylococcus*; *Str*.: *Streptococcus.*

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
