# Peer review of "Microbial Profile Antibacterial Properties and Chemical Composition of Raw Donkey Milk"

_animals, 2020, doi:10.3390/ani10112001_

Round 1

Reviewer 1 Report

Review:

Microbial composition and antibacterial properties of raw Donkey milk.

The authors describe the chemical and microbiological analysis of donkey milk. They have used state of the art methods, and managed to deepen the knowledge on the microbiological composition. While it was up to now accepted that donkey milk had antimicrobial activity, this work shows that this cannot be accepted as general rule, and advise that the milk should be pasturised before consumed. As such the document contains new data which is publishable.

The document is well written and easy to read and follow. Very few linguistic and text errors were noted, for which corrections are suggested below.

The rows of the text were not numbered, which complicates the editing.

  • The title should also include chemical composition, because it was carried out and brought in context with the microbiological data.
  • Is the name of the last author correct?

  • The text of the simple summary should be improved. Suggestion as follows:

 Apart from the economically exploited ruminants, alternative milk producing species, such as donkey, camel, reindeer, llama and yak have made an economic impact in many countries. However, their potential in human health and nutrition remains underexploited. In Europe, donkey milk is mainly produced in small-scale farms. However, consumer interest in donkey milk increases and is gaining international acceptance. This product is used as a substitute for human nutrition and susceptible consumer groups (i.e. infants and children who suffer from cow’s milk protein intolerance). The current study examines the donkey milk microbiota. In more detail, the findings have shown, the presence of lactic acid bacteria (LAB) in donkey milk, which is an important source for selecting starter cultures. Some pathogenic bacteria have also been identified. Further on that context, the European Union has adopted regulations (852/2004/EC) for donkey milk consumption, in order to achieve a high level of protection in human health. The conducted research focuses on the microbiological profile, antimicrobial properties and chemical composition of raw donkey milk, in order to provide useful information and improve expertise on the nutritional value and hygiene quality of donkey milk, of two representative Greek breeds.

  • In the abstract and key words 16s DNA is mentioned. Should this be 16s RNA?
  • Throughout the document the species names are not consistently used as full names or abbreviated names. Please correct throughout.
  • Materials and methods: the first 3 lines are in two letter sizes.
  • Further down in line 9 it is mentioned that the milk was refrigerated until analysed. What was the maximum time refrigerated?

Reviewer 2 Report

1.For lactococci growth temperature is 30oC. Please change!

Identification and technological properties of lactic acid bacteria isolated from raw goat milk of four Algerian races, Food Microbiology, Volume 21, Issue 5, 2004,Pages 579-588,

2.The LAB strains exhibited good technological performances, demonstrating that donkey milk is an interesting source of selecting starter cultures for dairy industry....

Did you perform any test regarding technological properties for the strains isolated apart from microbiological activity? Please specify!

Reviewer 3 Report

all my comment are attached on the pdf file

Round 2

Reviewer 3 Report

to my opinion new version is ok

Author Response

thank you